# Divergent Crosstalk Between Microglia and T Cells in Brain Cancers: Implications for Novel Therapeutic Strategies

**DOI:** 10.3390/biomedicines13010216

**Published:** 2025-01-16

**Authors:** Min-Hee Yi, Jinkyung Lee, Subin Moon, EunA So, Geonhyeok Bang, Kyung-Sub Moon, Kyung-Hwa Lee

**Affiliations:** 1Department of Microbiology and Immunology, Chonnam National University Medical School, Hwasun 58128, Jeollanam-do, Republic of Korea; minheeyi426@jnu.ac.kr (M.-H.Y.);; 2Biomedical Sciences Graduate Program (BMSGP), Chonnam National University, Hwasun 58128, Jeollanam-do, Republic of Korea; 3Department of Medicine, College of Medicine, Chosun University, Gwangju 61452, Republic of Korea; 4Department of Neurosurgery, Chonnam National University Hwasun Hospital, Hwasun 58128, Jeollanam-do, Republic of Korea; moonks@jnu.ac.kr; 5Department of Pathology, Chonnam National University Hwasun Hospital, Hwasun 58128, Jeollanam-do, Republic of Korea

**Keywords:** brain metastasis, glioblastoma, tumor-associated microglia, tumor-infiltrating T cells, tumor microenvironment

## Abstract

**Background**: Brain cancers represent a formidable oncological challenge characterized by their aggressive nature and resistance to conventional therapeutic interventions. The tumor microenvironment has emerged as a critical determinant of tumor progression and treatment efficacy. Within this complex ecosystem, microglia and macrophages play fundamental roles, forming intricate networks with peripheral immune cell populations, particularly T cells. The precise mechanisms underlying microglial interactions with T cells and their contributions to immunosuppression remain incompletely understood. **Methods**: This review comprehensively examines the complex cellular dialogue between microglia and T cells in two prominent brain malignancies: primary glioblastoma and secondary brain metastases. **Results**: Through a comprehensive review of the current scientific literature, we explore the nuanced mechanisms through which microglial-T cell interactions modulate tumor growth and immune responses. **Conclusions**: Our analysis seeks to unravel the cellular communication pathways that potentially underpin tumor progression, with the ultimate goal of illuminating novel therapeutic strategies for brain cancer intervention.

## 1. Introduction

Glioblastoma (GBM) and brain metastases (BM) represent some of the most aggressive forms of brain cancer, characterized by profound therapeutic resistance and poor clinical outcomes [1,2,3]. Despite current standard treatments, including surgical intervention followed by chemotherapy and radiation therapy, patients with GBM experience severely limited survival, typically around 20 months [4]. Patients diagnosed with lung cancer with BM face an even more dire prognosis with a remarkably low one-year survival rate ranging from 4% to 20% [5]. Consequently, comprehending the mechanisms driving brain tumor occurrence and progression is paramount for identifying potential therapeutic interventions.

Mounting evidence increasingly highlights the critical role of the tumor microenvironment (TME) in driving tumor progression. The TME significantly facilitates GBM advancement and confers resistance to chemotherapy and immunotherapy [6,7]. Morphological activation of microglia, coupled with a substantial macrophage presence near brain tumors, underscores their profound involvement in the TME [8]. Recent research emphasizes the collaborative function of tumor-associated microglia (TAMs) with peripheral immune cells, particularly T cells, and macrophages, which constitute the predominant immune cell population within TME components [9,10].

Microglia, the brain’s resident immune cells, traditionally support and safeguard neuronal function within the central nervous system (CNS) [11,12,13]. Emerging studies have increasingly highlighted their significant involvement in brain tumor development [14]. Their anti-inflammatory actions facilitate tumor cell survival and colonization within the brain parenchyma [15]. As primary representatives of the innate immune system in the CNS, brain-resident microglia are crucial for immune surveillance and host defense [16,17]. Beyond neuroinflammatory roles, microglia also participate in critical processes such as synapse pruning and remodeling [18].

Under normal conditions, microglia maintain a resting state, expressing minimal levels of major histocompatibility complex (MHC) class I and class II molecules, including CD86 and CD40 [19,20,21,22]. However, they demonstrate responsiveness to microenvironmental changes by presenting their remarkable processes [11,12,13]. Upon CNS injury, microglia undergo activation characterized by increased proliferation, motility, phagocytic activity, and release of inflammatory cytokines and reactive oxygen species [23,24]. Functioning as antigen-presenting cells (APCs), activated microglia closely resemble peripheral macrophages and play pivotal roles in both innate and adaptive immune responses. They elevate the expression of MHC and costimulatory molecules, contributing to CD4- and CD8-specific T-cell responses, thereby protecting against invading pathogens and promoting healing [25,26,27].

Nonetheless, substantial debate persists regarding microglia’s interaction with peripheral immune cells, their function in immune suppression within the TME, and their contribution to brain cancer progression. This review discusses the mechanisms underlying immunosuppression and intratumoral immunity emerging from the intricate interplay between microglia and T cells in both GBM and BM. Drawing from recent advancements in understanding microglia’s role in CNS immunity, we explore how their collaboration influences tumor growth, TME dynamics, and anti-tumor immune responses. Additionally, we propose cellular mechanisms to elucidate the complex interaction between microglia and T cells.

## 2. Comparative Overview of Glioblastoma and Brain Metastases

GBM represents the most common and aggressive primary brain tumor, constituting approximately 45% of malignant brain tumors in adult populations [28]. GBM can arise de novo, primarily originating from glial cells in the brain, or potentially evolve from low-grade gliomas. The tumor is characterized by its remarkable rapid growth and highly invasive nature [29]. Pathologically, GBM demonstrates distinctive molecular features, including overexpression of vascular endothelial growth factor (VEGF), which critically promotes angiogenesis and directly correlates with tumor malignancy and clinical prognosis [30]. Genetically, GBM frequently exhibits complex mutations across multiple crucial genes, including *TP53*, *PTEN*, and *EGFR*. This genetic landscape contributes to the tumor’s profound intratumoral heterogeneity [31,32]. The molecular complexity of GBM underscores its aggressive biological behavior and resistance to conventional treatment modalities.

BM represents secondary brain tumors that originate from primary malignancies elsewhere in the body, most commonly emerging from lung and breast cancers and melanoma [3,33,34,35]. Unlike primary brain tumors, BM’s genetic profiles intrinsically reflect the mutations present in their original tumor sites. Lung cancer BM, for instance, frequently retains genetic alterations characteristic of the primary lung cancer, such as *EGFR* mutations and *ALK* rearrangements observed in non-small cell lung cancer [36].

Clinically, GBM and BM demonstrate distinct pathological behaviors and symptomatic presentations. GBM progresses with exceptional rapidity, manifesting through severe neurological symptoms including headaches, seizures, cognitive and personality changes, and neurological deficits [37]. Despite implementing aggressive multimodal treatment strategies involving surgical intervention, radiation therapy, and chemotherapy, the median patient survival remains limited to 12–18 months due to the tumor’s inherently aggressive biological characteristics [38]. BM similarly presents with variable symptomatic manifestations depending on the specific tumor location, typically characterized by headaches and focal neurological impairments [39]. The clinical course of BM is significantly influenced by the management of the primary cancer and systemic metastatic lesions. Treatment approaches are intricately tailored to address the specific characteristics and molecular profile of the primary malignancy.

## 3. Microglia in Glioblastoma and Brain Metastases

### 3.1. Molecular Mechanisms of Microglial Involvement in GBM Progression

Microglia, the resident immune cells of the CNS, play pivotal roles in diverse physiological and pathological processes. These cells are fundamental to brain development, engaging in dynamic interactions with neurons and glial cells, maintaining neuronal synapses, and coordinating tissue repair mechanisms in response to damage [40,41]. In GBM, microglia demonstrate a complex and nuanced functional profile, simultaneously acting as potential tumor suppressors and promoters contingent upon specific contextual conditions.

During the early stages of tumor development, microglia can function as anti-tumor agents by actively recognizing and targeting nascent tumor cells. They prevent tumor growth through the secretion of pro-inflammatory cytokines known to inhibit tumor progression, including tumor necrosis factor-alpha (TNF-α), interleukin-1β (IL-1β), IL-6, IL-8, IL-12, and IL-23 [42]. One recent study has explored the intricate immunomodulatory effects of miRNA-125b expression and STAT3 signaling during microglial interactions with GBM [43]. Notably, reduced expression of oncogenic miRNA-125b appears to shift the inflammatory profile of microglia towards a more active anti-tumor phenotype [43]. Additionally, GBM-associated microglia-derived exosome circKIF18A has been observed to promote angiogenesis by targeting the FOXC2 molecular pathway [44].

As tumor progression advances, microglia frequently transit toward a tumor-promoting role [45]. This transformation is particularly evident in their shift to an alternative microglial state that facilitates pro-tumorigenic functions through the secretion of anti-inflammatory factors such as transforming growth factor-beta (TGF-β) [46,47]. In patients with higher-grade gliomas, a significant increase in CD204^+^ TAM has been documented [48,49].

The anti-inflammatory state of microglia substantially supports tumor growth by releasing cytokines and growth factors that stimulate critical processes including angiogenesis, cancer cell proliferation, and the creation of an immunosuppressive TME. This environment effectively enables tumor cells to evade immune detection [50,51,52,53]. Microglia’s tumor-promoting activities are further enhanced by their anti-inflammatory and homeostatic functions. Under these circumstances, these cells may prioritize tissue homeostasis over mounting an effective anti-tumor response, thereby unintentionally supporting tumor progression.

Simultaneously, microglia frequently exist in an activated state characterized by the upregulation of pro-inflammatory cytokines, chemokines, and growth factors that promote tumor growth and angiogenesis, though this activation state demonstrates contextual variability [54,55,56,57]. The pro-inflammatory status of microglia involves complex signaling pathways such as the Toll-like receptor 2 (TLR2), chemokine receptor 1 (CCR1), CXCR4-STAT3 axis, and CXCR4-mediated infiltration of pro-tumoral myeloid cells [58,59,60,61,62], and colony-stimulating factor-1 receptor (CSF-1R) pathways [54,55,56,63]. Notably, inhibition of CSF-1R has demonstrated the potential to modify microglial activation and suppress glioma progression, underscoring the critical role of these cells in the GBM microenvironment [55].

Understanding the intricate inflammatory and homeostatic functions of microglia within the TME is paramount. The emerging evidence suggests that rather than actively combating tumor cells, microglia may inadvertently create a supportive environment for tumor growth. Consequently, exploring innovative therapeutic strategies to reprogram microglia from tumor-supporting to tumor-combating states represents a critical avenue of research. Targeting the signaling pathways that regulate microglial alternative and anti-inflammatory activities could potentially disrupt their pro-tumor functions and restore their tumor-suppressive capabilities, offering promising new approaches to treating aggressive brain cancers, including GBM and BM (Table 1).

### 3.2. Molecular Mechanisms of Microglial Involvement in BM Progression

Microglia demonstrate extraordinary cellular plasticity, enabling them to adopt diverse activation states in response to various stimuli and pathological events, particularly in BM [64,65,66,67,68]. Within the context of BM, microglia can transit between activation states based on complex cues from tumors and their microenvironments [68,69], generating intricate interactions that influence disease progression.

BM emerges when cancer cells disseminate from their primary site to the brain, generating secondary tumors [70]. This metastatic process is facilitated by blood–brain barrier disruption, which allows circulating tumor cells to invade the CNS [71,72]. Microglia function as primary responders to these infiltrating cancer cells, exhibiting remarkable responsive capabilities [64,67,68].

Microglial activation exists on a complex spectrum rather than in discrete categories, with pro-inflammatory and anti-inflammatory states representing dynamic extremes [73]. In their pro-inflammatory state, microglia generate cytokines, reactive oxygen species, and molecular compounds capable of inhibiting tumor growth by promoting anti-tumor immune responses. Conversely, during anti-inflammatory states, microglia release cytokines, tissue remodeling factors, and angiogenic molecules that can support tumor growth and metastasis [74].

Multiple molecular mechanisms regulate microglial activation states in BM. These cells participate in local inflammatory responses and interact with metastatic cells through sophisticated signaling pathways, such as CXCL12-CXCR4 [61]. Such interactions can generate factors supporting tumor growth and invasion, including nitric oxide synthase (NOS) and cyclooxygenase-2 (COX-2) [75,76]. One recent study has revealed that heat shock protein 47 (HSP47) is overexpressed in BM, with elevated levels predicting poor patient survival [53]. Critical signaling pathways, including NF-κB, STAT3, and PI3K/Akt, can be activated by tumor-derived signaling mechanisms [62,77].

Emerging research suggests that inducing microglia into an inflammatory state can enhance radiotherapy effectiveness and achieve anti-tumor effects, ultimately suppressing tumor growth [78]. Blocking specific molecular axes, such as the macrophage migration inhibitory factor (MIF)/CD74 pathway, can polarize microglial cells into a pro-inflammatory state, potentially inhibiting tumor progression after radiation treatment [78]. Microglia express pattern recognition receptors (PRRs) like Toll-like receptors, which detect damage-associated molecular patterns (DAMPs) released by tumor cells and initiate inflammatory responses [79]. Epigenetic modifications, including DNA methylation and histone acetylation, also play crucial roles in shaping microglia phenotype and functional characteristics. Recent studies have increasingly highlighted the significance of microRNAs and long non-coding RNAs in regulating microglia gene expression in the TME [80]. Understanding the nuanced characteristics and roles of microglia in brain metastases is essential for developing targeted therapeutic strategies that can modulate their function and potentially improve patient outcomes (Table 2).

### 3.3. Microglial Dynamics in Brain Tumor Microenvironments

Recent investigations have increasingly illuminated the substantial involvement of immune cells in brain tumor development [14]. Anti-inflammatory actions within the brain parenchyma of the CNS facilitate tumor cell survival and colonization [14,81]. As primary representatives of the innate immune system in the CNS, brain-resident microglia execute critical roles in immune surveillance and host defense [16,17].

Advanced RNA sequencing technology has enabled the sophisticated identification of diverse immune cell populations within the TME. A recent study has characterized myeloid cells with distinct expression profiles from CD11b^+^ cells in murine orthotopic GL261 glioma models [82]. Notably, researchers observed upregulation of major histocompatibility complex (MHC) class I and II molecules, alongside increased expression of Bst2 and Lgals3bp, characteristics associated with disease-related microglia in tumor-bearing hemispheres. Additionally, heightened CCL2 expression, critical for CCR2^+^ monocyte recruitment, was consistently detected [82,83].

Microglia within the TME demonstrated elevated expression of proliferation-associated genes, including *Stmn1*, *Tubb5*, *Tuba1b*, *Cdk1*, and *Top2a* [82,84,85]. Furthermore, distinctive upregulation of proteolytic regulatory genes such as *Timp2*, *Serpine2*, *Cst7*, and *Ctsd* suggests potential supportive roles in tumor invasion through extracellular matrix (ECM) reorganization. Dynamic cellular changes in GBM revealed complex monocyte/macrophage populations, including inflammatory monocytes and (Ly6c2^high^, Ccr2^high^, and Tgfbi^low^), an intermediate state of monocytes and macrophages (Ly6c2^high^, Ccr2^high^, and Tgfb^high^), and differentiated macrophages (Ly6c2^high^, Ifitm2^high^, Ifitm3^high^, and S100ab^high^) [82]. The identification of precise cellular markers has become crucial for comprehensive microglial analysis. Researchers have proposed Tmem119 as a marker for microglia during CNS inflammation following nerve injury [86], while Galectin-3 (Gal-3) serves as a critical marker for macrophages engaged in tumor immunosuppression [82].

Recent advancements in imaging mass cytometry have provided spatial insights into primary and metastatic brain tumors, significantly enhancing our understanding of myeloid cell populations. Notably, a unique population of myeloperoxidase (MPO)-positive macrophages has been associated with long-term survival in human GBM. These MPO+ macrophages exhibit a pro-inflammatory, anti-tumorigenic phenotype, expressing markers like S100A8 and S100A9, with signatures linked to reactive oxygen species biosynthesis and HIF1α signaling [87]. High MPO^+^ macrophage density correlates with increased innate effector responses, reduced immunosuppressive signaling, and improved patient survival, underscoring their potential as favorable contributors to the tumor microenvironment [87].

Single-cell RNA sequencing (scRNA-seq) analysis has revealed significant limitations in mouse models’ ability to fully represent the functional heterogeneity observed in tumor-associated macrophages from patients with GBM [55,88]. Human GBM studies have characterized novel pro-inflammatory and proliferative microglial populations [89] and immunosuppressive CD163^+^HMOX1^+^ microglia capable of inducing T cell exhaustion through interleukin-10 (IL-10) release [90]. HMOX1^+^ microglia strategically position themselves at the interface between GBM cells and T cells, facilitating T cell exhaustion mechanisms [90]. The research underscores the complex contributions of multiple tumor-associated microglia subpopulations in mediating immune evasion within GBM.

Comparative genetic analyses have unveiled intriguing distinctions between mouse and human tumor microenvironments. Specific genes such as *Cst*, *Hexb*, and *Sparc* exhibit significant differential expression between microglia and macrophages in mouse tumors but not in human tumors. Some markers, including APOC2, TMIGD3, and SCIN are exclusively characteristic of human tumor microglia [55]. Particularly in human GBM specimens, MARCO^hi^ macrophages and CD163^+^HMOX1^+^ microglia have been exclusively identified in mesenchymal GBM subtypes [90,91]. These findings highlight the intricate cellular dynamics and molecular mechanisms underlying microglial interactions in brain tumor environments, emphasizing the critical need for continued investigation into these complex immunological processes.

## 4. Crosstalk Between Microglia and T Cells in Glioblastoma

### 4.1. Mechanisms of Microglia-T Cell Interactions in GBM

In response to the tumor presence, diverse immune cells are recruited to the TME [40], engaging in complex interactions with T cells that can either suppress tumor growth or support immune evasion [92,93]. Recent studies reveal that microglial suppression of T cells can potentially enhance GBM growth [94,95], while augmenting T cell anti-tumor activity may inhibit tumor progression [96].

Different T cell subpopulations interact with microglia through unique molecular mechanisms [97,98,99,100], significantly impacting tumor progression [10]. CD8+ cytotoxic T cells can be either activated or suppressed depending on tumor antigen presentation, particularly by microglia, as evidenced in recent studies [101]. In glioblastoma (GBM), tumor-associated macrophages (TAMs), including microglia, frequently express elevated levels of HLA class I molecules, which engage with CD8 receptors on T cells. This interaction plays a pivotal role in enabling T cells to recognize tumor antigens and drive their activation [95,101,102]. CD4+ helper T cells interact with microglia to induce either Th1 or Th2 responses, thereby modulating anti-tumor immunity or contributing to an immunosuppressive microenvironment [103,104,105,106]. Recent research highlights that the tumor microenvironment in glioblastoma (GBM) can significantly influence the polarization of these responses, offering critical insights for developing effective treatment strategies [107].

Regulatory T cells (Tregs) can be activated by microglia to facilitate tumor immune evasion, a phenomenon frequently observed in the glioblastoma (GBM) microenvironment, where Tregs exhibit excessive activation that supports tumor progression [106,108,109]. Recent studies have further identified markers of T cell exhaustion in GBM, such as elevated expression of ICOS, CTLA4, TIGIT, IL2RA, and IL10RA in Tregs, highlighting their role in facilitating tumor growth [101]. Therefore, within the GBM microenvironment, Tregs frequently demonstrate excessive activation, supporting tumor growth [110]. Gamma delta T cells (γδ T cells) interact with microglia to regulate both innate and adaptive immune responses [111], playing crucial roles in GBM progression through intricate interactions [10,112,113,114].

### 4.2. Impact of Microglia-T Cell Interactions on Tumor Progression and Immune Evasion

Microglia play a dual role in glioblastoma (GBM) by modulating immune responses and contributing to immune suppression within the tumor microenvironment (TME). Microglia recognize abnormal protein fragments produced by the tumor cells and present these antigens to T cells, primarily activating CD8^+^ cytotoxic T cells to target tumor cells [95,115]. However, microglia also secrete critical cytokines such as interferon-γ (IFN- γ), IL-1, and IL-6, alongside chemokines like CCL2 and CXCL10, which modulate T cell activity and recruit immune cells to the tumor site [95,116,117,118]. The interferon gamma receptor (IFNGR) signaling pathway has been identified as pivotal for GBM susceptibility to CAR-T cell immunotherapy, enabling effective tumor cell elimination in solid tumors [119].

Despite this, microglia contribute to immune suppression within the TME. They express programmed death-ligand 1 (PD-L1) [120], which interacts with the PD-1 receptor on T cells, effectively inhibiting their ability to attack tumor cells [121]. Direct cell-to-cell interactions between microglia and T cells involve complex antigen presentation and signaling pathway mechanisms [122,123]. These interactions, along with microglial secretion of metabolic byproducts like lactate, suppress T cell activity and support tumor immune evasion [122,123]. These mechanisms enable tumors to evade immune detection and proliferate. Recent studies suggest that PD-1 immune checkpoint inhibitors can disrupt this immunosuppressive interaction, potentially restoring T cell’s anti-cancer function [124,125]. Additionally, a gene called TIGIT, highly expressed in CAR-T cells from non-responsive patients, has been shown to induce exhaustion and dysfunction in these cells, posing a significant challenge to immunotherapy efficacy [126].

### 4.3. Molecular Signaling Pathways

The nuclear factor kappa B (NF-κB) pathway plays a significant role in inducing anti-inflammatory responses and regulating immune reactions [127,128]. The phosphatidylinositol 3-kinase/Akt/mammalian target of the rapamycin (PI3K/Akt/mTOR) pathway, governing cell survival and proliferation, can support tumor progression [129].

Understanding these intricate cytokines, chemokines, and signaling pathways is crucial for advancing targeted immunotherapeutic strategies in brain cancers. These complex interactions between microglia and T cells represent critical mechanisms facilitating tumor growth and immune evasion, presenting promising targets for therapeutic intervention. Further investigation into microglia-T cell interactions within the TME remains essential for developing novel strategies to counter tumor immune evasion and enhance immunotherapy effectiveness (Table 3).

## 5. Crosstalk Between Microglia and T Cells in Brain Metastasis

The complex interactions between microglia and T cells serve as critical mediators of tumor growth and immune evasion in BM [81,132]. These interactions manifest differently across various cancer types. In lung cancer BM, microglia can express PD-L1, which binds to PD-1 on T cells and suppresses their activity, thereby promoting tumor progression [133,134]. For breast cancer BM, microglia facilitate immune evasion by activating regulatory T cells (Tregs), which subsequently inhibit cytotoxic T cell function [68,135]. In melanoma BM, microglia engage with γδ T cells to modulate both innate and adaptive immunity, though these interactions vary depending on the cancer type and metastatic progression [136,137,138]. Microglia further support tumor progression through matrix metalloproteinase (MMPs) secretion [139], which aids cancer invasion, and VEGF production [140,141,142], which promotes angiogenesis.

The immunosuppressive mechanisms driven by microglia encompass multiple pathways that create a tumor-permissive microenvironment. Through the secretion of immunosuppressive cytokines such as TGF-β and IL-10, microglia effectively suppress cytotoxic T-cell activity and dampen inflammatory responses [143,144,145]. Microglia also release specific chemokines, including CCL2 [146] and CXCL10 [147], which attract immune cells to the tumor site, though these chemokines may paradoxically contribute to an immunosuppressive tumor microenvironment under certain conditions. The expression of PD-L1 by microglia serves as another crucial mechanism for T cell inhibition, representing a key target for immune checkpoint therapies [148,149]. Through these various mechanisms, particularly the support of angiogenesis via VEGF secretion and promotion of regulatory immune mechanisms, microglia significantly enhance tumor survival and growth potential [150].

Understanding the signaling pathways and molecular interactions has become essential for therapeutic development. The PD-1/PD-L1 pathway, particularly prominent in lung cancer brain metastases, functions as a key mechanism of T cell suppression and immune evasion. In breast cancer brain metastases, PI3K/Akt/mTOR pathway promotes tumor growth and therapeutic resistance [151,152]. The JAK/STAT pathway in melanoma metastases plays a crucial role in regulating immune cell activity and supporting immune evasion [153]. Future immunotherapeutic strategies may achieve improved treatment outcomes for brain metastasis by targeting these pathways and disrupting interactions between microglia and T cells (Table 4). The complex interactions between microglia and the glioblastoma/brain metastasis microenvironment are illustrated in Figure 1.

## 6. Therapeutic Implications and Future Directions

The interactions between microglia and tumor cells exhibit dynamic and multifaceted characteristics. Tumor cells modulate microglial activation and function through the secretion of various factors, including cytokines, chemokines, and extracellular vesicles. These tumor-derived factors can drive microglia towards a state that promotes a pro-tumorigenic environment, thereby facilitating tumor cell survival, proliferation, and invasion. In BM, microglia participate in pre-metastatic niche formation through multiple mechanisms, including extracellular matrix alteration, angiogenesis enhancement, and local immune response suppression. Notably, microglia can also demonstrate anti-tumor effects under specific conditions by recognizing and phagocytosing tumor cells, producing cytotoxic molecules, and recruiting other immune cells to the tumor site. The metastatic process in the brain is critically determined by the balance between these pro- and anti-tumor activities of microglia.

The therapeutic targeting of microglia in brain metastases presents both significant challenges and opportunities for intervention. Strategies aimed at reprogramming microglia from a pro-tumorigenic to an anti-tumorigenic state show promise for enhancing anti-tumor immunity and inhibiting tumor progression. These potential approaches include the utilization of small molecules, antibodies, or gene therapy to modulate key signaling pathways and epigenetic regulators of microglia activation. Furthermore, combination therapies targeting both microglia and other components of the TME may provide synergistic effects and improve treatment efficacy.

Additional research remains necessary to fully elucidate the complex interactions between microglia, tumor cells, and immune cells, as well as to develop safe and effective therapies that can selectively target microglia while preserving their essential homeostatic functions in the TME. The continuing expansion of knowledge regarding microglia biology and their role in brain cancers suggests the likely emergence of new therapeutic avenues, offering hope for patients with brain cancer with dismal prognoses.

Recently, oncolytic viruses have emerged as a promising therapeutic approach for glioblastoma, with numerous clinical trials focusing on their application [154,155,156]. Oncolytic virotherapy has demonstrated the potential to modulate immune cell interactions within the tumor microenvironment (TME), as shown in single-cell analysis studies [157]. Notably, oncolytic viruses enhance immune checkpoint interactions between macrophages and CD8^+^ T cells, as observed in malignant ascites, where virotherapy improved immune crosstalk and promoted tumor-specific cytotoxicity. These findings underline the relevance of targeting similar crosstalk mechanisms in glioblastoma, particularly between microglia and T cells, to achieve therapeutic benefit.

Therefore, therapeutic strategies targeting microglia-T cell interactions represent a promising avenue for glioblastoma treatment. Oncolytic viruses, already extensively studied in clinical trials for glioblastoma, offer a unique opportunity to reprogram microglia from a pro-tumorigenic to an anti-tumorigenic state. By enhancing microglia-T cell interactions and promoting tumor-specific immune responses, oncolytic virotherapy could overcome glioblastoma’s highly immunosuppressive TME. The ability of oncolytic virotherapy to induce specific T cell subsets, such as CXCR6^+^ and GZMK^+^ CD8^+^ T cells, and to potentiate immune checkpoint therapies highlights its potential for combination treatments that target microglia and T cells concurrently.

Further research is essential to unravel the complexities of microglial interactions with tumor and immune cells and to develop innovative strategies that enhance microglia-T cell crosstalk within the tumor microenvironment. Understanding these intricate interactions will enable the design of safe and effective therapies that selectively target microglia while preserving their essential homeostatic functions. While optimizing the delivery and specificity of oncolytic virotherapy remains a promising avenue, additional efforts are needed to explore alternative approaches such as small molecules, gene editing, and immune-modulating therapies to reprogram microglia and strengthen their interactions with T cells. The continued expansion of knowledge regarding microglia biology and their role in brain cancers offers new opportunities for developing therapies that enhance immune cell crosstalk. Whether through oncolytic viruses or other innovative strategies, such treatments have the potential to significantly improve outcomes for patients with glioblastoma and bring hope to those facing a poor prognosis.

## 7. Conclusions

The comprehensive understanding of the divergent crosstalk between microglia and T cells in GBM and BM remains crucial for developing targeted therapies. Through the elucidation of mechanisms underlying these interactions, researchers can identify novel therapeutic targets and strategies to improve treatment outcomes for patients affected by these devastating brain tumors.

## Figures and Tables

**Figure 1 biomedicines-13-00216-f001:**
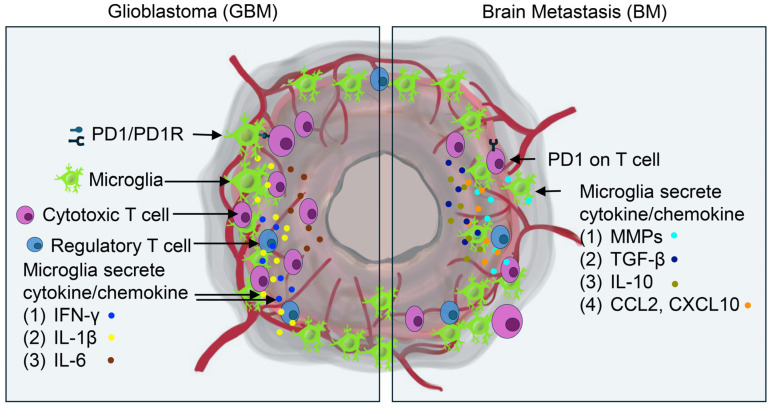
Microglial interactions within the tumor microenvironment of glioblastoma and brain metastases. Schematic representation of bidirectional signaling between microglia and various cellular components of the tumor microenvironment. Key cytokine networks and molecular mediators secreted by activated microglia are shown, highlighting their roles in immunomodulation, tumor progression, and maintenance of the tumor microenvironment.

**Table 1 biomedicines-13-00216-t001:** Molecular Mechanisms of Microglial Involvement in GBM.

Mechanism	Description	References
Pro-Inflammatory Mechanisms	Cytokine secretion: TNF-α, IL-1β, IL-6, IL-8, IL-12, and IL-23 inhibit tumor progression.	[42]
	miRNA-125b downregulation: Shifts microglial profile to an anti-tumor state.	[43]
	Pathways: TLR2, CCR1, and CXCR4-STAT3 axis mediates activation.	[54,55,56,58,59,60,61,62,63]
Anti-Inflammatory Mechanisms	Cytokine secretion: TGF-β supports tumor growth and immune evasion.	[46,47]
	Transition to an alternative microglial state promotes angiogenesis, tumor proliferation, and immunosuppressive TME.	[50,51,52,53]
	Increased CD204^+^ TAMs in high-grade gliomas.	[48,49]
Microglial Plasticity	Microglia display context-dependent functions. Early-stage GBM: Anti-tumor role via pro-inflammatory cytokines and targeting nascent tumor cells.	[42]
	Advanced-stage GBM: Transition to tumor-promoting roles due to anti-inflammatory and homeostatic responses.	[50,51,52,53]
Angiogenesis Promotion	Exosome circKIF18A: Drives angiogenesis by targeting FOXC2 pathways. Microglia release growth factors supporting vascularization of the tumor environment.	[44]
Therapeutic Targeting	CSF-1R inhibition: Suppresses glioma progression and modifies microglial activation.	[55]
	Reprogramming microglia to anti-tumor states by targeting pro-tumor signaling pathways (e.g., CXCR4, STAT3) offers potential therapeutic strategies.	[43,55]

**Table 2 biomedicines-13-00216-t002:** Molecular Mechanisms of Microglial Involvement in BM Progression.

Mechanism	Description	References
Pro-Inflammatory Mechanisms	Microglia produce cytokines, reactive oxygen species, and molecular compounds that promote anti-tumor immune responses.	[73]
	Activation of inflammatory pathways such as NF-κB and STAT3 enhances tumor suppression potential.	[62,77]
Anti-Inflammatory Mechanisms	Anti-inflammatory microglia secrete cytokines, tissue remodeling factors, and angiogenic molecules that support tumor growth and metastasis.	[74]
	Tumor-derived signaling pathways (e.g., CXCL12-CXCR4) promote tumor invasion and survival.	[61,75,76]
Microglial Plasticity	Microglia exhibit cellular plasticity, transitioning between pro-inflammatory and anti-inflammatory states based on tumor and microenvironmental cues.	[64,65,66,67,68,69]
	BM occurs when cancer cells metastasize to the brain via disrupted blood–brain barrier.	[70,71,72]
Epigenetic and RNA Regulation	Epigenetic modifications like DNA methylation and histone acetylation shape microglial phenotypes. MicroRNAs and long non-coding RNAs regulate microglial gene expression in the TME.	[80]
Therapeutic Targeting	Inducing pro-inflammatory microglia enhances radiotherapy effectiveness and suppresses tumor growth. Blocking MIF/CD74 pathway polarizes microglia to pro-inflammatory states, inhibiting tumor progression post-radiotherapy.	[78]
	PRRs like TLRs detect DAMPs to initiate inflammatory responses.	[79]

**Table 3 biomedicines-13-00216-t003:** Key Mechanisms in Microglia-T Cell Interactions in GBM.

Mechanism	Description	References
Antigen Presentation	Microglia present tumor-derived antigens to T cells, influencing their activation status (e.g., CD8+ cytotoxic T cells).	[95,115]
Cytokine Secretion	Secretion of cytokines like IFN-γ, IL-1, and IL-6 modulates T cell activity and shapes immune responses.	[116,117]
Chemokine Release	Chemokines such as CCL2 and CXCL10 recruit T cells to the tumor microenvironment.	[95,130]
Immune Evasion Pathways	Microglia express PD-L1, interacting with PD-1 receptors on T cells to suppress their activity.	[120,121]
Regulatory T Cell Activation	Microglia activate Tregs, which suppress cytotoxic T cells and enhance immune evasion.	[104,131]
Metabolic Byproducts	The release of lactate and other byproducts suppresses T cell function, aiding tumor immune evasion.	[122,123]

**Table 4 biomedicines-13-00216-t004:** Key Mechanisms in Microglia-T Cell Interactions in BM.

Cancer Type	Microglia-T Cell Interaction	Key Mechanisms	Pathways Involved	References
Lung Cancer BM	Microglia express PD-L1, binding to PD-1 on T cells, suppressing activity and promoting tumor progression.	- T cell inhibition via PD-L1 expression. - Promotion of immune evasion.	PD-1/PD-L1	[133,134]
Breast Cancer BM	Microglia activate regulatory T cells (Tregs), inhibiting cytotoxic T cells and facilitating immune evasion.	- T cell suppression by Tregs. - Tumor growth and resistance via PI3K/Akt/mTOR pathway.	PI3K/Akt/mTOR	[68,135,151]
Melanoma BM	Microglia interact with γδ T cells, modulating innate and adaptive immunity.	- Immune cell regulation via JAK/STAT pathway. - Angiogenesis via VEGF secretion.	JAK/STAT	[136,138,153]
General Mechanism	- Microglia secrete MMPs for invasion. - VEGF promotes angiogenesis. - Immunosuppressive cytokines like TGF-β and IL-10 suppress inflammation. - Chemokines (CCL2, CXCL10) attract immune cells but can aid immunosuppression. - PD-L1 expression inhibits T cells.	- Immune suppression via cytokines. - Tumor microenvironment modulation. - Enhanced survival and growth.	VEGF, TGF-β, IL-10, PD-1/PD-L1, CCL2, CXCL10	[144,147,150]

## Data Availability

Data sharing is not applicable to this article as no new data were created or analyzed in this study.

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
