# Peer review of "Divergent Crosstalk Between Microglia and T Cells in Brain Cancers: Implications for Novel Therapeutic Strategies"

_biomedicines, 2025, doi:10.3390/biomedicines13010216_

Round 1
Reviewer 1 Report
Comments and Suggestions for Authors
This review addresses the critical issue on the crosstalk between glioblastoma and T cells in both microglioma and brain metastasis. The manuscript is well-structured, covering the mechanistic role of microglia and microglia-T cell interactions in brain cancers. I recommend publication after the authors additionally address several major issues that I am concerned about.
1. Like macrophages, the microglial cell has a dual effect including both anti- and pro-inflammatory states in brain tumors. Are there any specific or representative markers that may define the different inflammatory state of microglial cells? for example, the M1 (MHC) and M2 (CD206) states in macrophages. This would be helpful to characterize the subtypes of microglial cells.
2. Regarding the comparative overview of glioblastoma and brain metastases. A recent article used cutting-edge imaging mass cytometry to reveal the spatial information of both primary and metastastic brain tumors (PMID: 36725935). Such study or any other similar studies should be included and summarized.
3. The “Therapeutic Implications and Future Directions” part is inadequate. In recent years, oncolytic virus has been intensively applied to treat glioblastoma through multiple clinical trials (PMID: 35864254, 37853118, 37188783, 35864115 etc.). Also, oncolytic virus has been shown to enhance cellular crosstalk, especially the immune checkpoint crosstalk between macrophages and T cells in malignant ascites as reported in a single-cell analysis (PMID: 38659226). Therefore, the future treatment prospects focusing on targeting crosstalk between microglia and T cells to treat glioblastoma should also summarized the current application and emerging potential of oncolytic virus.
4. A graphical illustration would be helpful to better describe the cellular interaction between microglial cell and T cell.
Author Response
Reviewer #1: This review addresses the critical issue on the crosstalk between glioblastoma and T cells in both microglioma and brain metastasis. The manuscript is well-structured, covering the mechanistic role of microglia and microglia-T cell interactions in brain cancers. I recommend publication after the authors additionally address several major issues that I am concerned about.
Comments 1: Like macrophages, the microglial cell has a dual effect including both anti- and pro-inflammatory states in brain tumors. Are there any specific or representative markers that may define the different inflammatory state of microglial cells? for example, the M1 (MHC) and M2 (CD206) states in macrophages. This would be helpful to characterize the subtypes of microglial cells
Response 1: Thank you for your insightful question. Similar to macrophages, microglial cells exhibit dual roles in brain tumors, characterized by pro-inflammatory (M1-like) and anti-inflammatory (M2-like) states. Representative markers for these states in microglia include: (1) M1-like (pro-inflammatory state): iNOS, TNF-α, IL-1β, and MHC-II, and (2) M2-like (anti-inflammatory state): Arg1, YM1, and TGF-β. While these markers can provide insights into the functional roles of microglia, it is important to recognize that their states are dynamic and do not always fit neatly into the M1 or M2 categories (PMID: 36327895).
Currently, instead of aiming for a perfect binary distinction, research is more focused on identifying the predominant inclination (pro-inflammatory or anti-inflammatory) of microglial cells. This can be inferred using the aforementioned inflammatory markers, which offer valuable clues for characterizing microglial subtypes in the tumor microenvironment (PMID: 36327895).
Comments 2: Regarding the comparative overview of glioblastoma and brain metastases. A recent article used cutting-edge imaging mass cytometry to reveal the spatial information of both primary and metastastic brain tumors (PMID: 36725935). Such study or any other similar studies should be included and summarized.
Response 2: Thank you for your valuable feedback. We recognize the importance of incorporating findings from imaging mass cytometry studies (PMID: 36725935), which offer spatial insights into primary and metastatic brain tumors. We would like to note that the key findings from this study, particularly the spatial characterization of myeloid cell populations and the identification of MPO-positive macrophages associated with long-term survival, have already been summarized and integrated into Section 3.3, Microglial dynamics in brain tumor microenvironments.
Comments 3: The “Therapeutic Implications and Future Directions” part is inadequate. In recent years, oncolytic virus has been intensively applied to treat glioblastoma through multiple clinical trials (PMID: 35864254, 37853118, 37188783, 35864115 etc.). Also, oncolytic virus has been shown to enhance cellular crosstalk, especially the immune checkpoint crosstalk between macrophages and T cells in malignant ascites as reported in a single-cell analysis (PMID: 38659226). Therefore, the future treatment prospects focusing on targeting crosstalk between microglia and T cells to treat glioblastoma should also summarized the current application and emerging potential of oncolytic virus.
Response 3: We thank you for your valuable comment and for highlighting the importance of targeting microglia-T cell crosstalk in therapeutic strategies. In response to your suggestion, we have incorporated the referenced research and revised the Therapeutic Implications and Future Directions section to include the current application of oncolytic viruses in glioblastoma treatment, as well as their potential to enhance cellular crosstalk, particularly immune checkpoint interactions between macrophages and T cells, as reported in recent studies. These updates emphasize the need for developing therapies that specifically target microglia-T cell interactions, underscoring their critical role in modulating the tumor microenvironment and improving treatment outcomes in glioblastoma.
Comments 4: A graphical illustration would be helpful to better describe the cellular interaction between microglial cell and T cell.
Response 4: We have added a graphical illustration to the manuscript to better depict the cellular interactions between microglial cells and T cells. Thank you for your insightful suggestion!
Reviewer 2 Report
Comments and Suggestions for Authors
I carefully evaluated the manuscript entitled “Divergent Crosstalk Between Microglia and T cells in Brain Cancers: Implications for Novel Therapeutic Strategies”. This paper can’t be accepted to publish in MDPI biomedines journal. Reasons for this decision:
Ø The theme of this review paper “implications for novel therapeutic strategies” is missing. Authors are requested to discuss more on therapeutic implications.
Ø Authors are requested to add some infographics representing the theme of this review.
Ø Found similar review with title “Crosstalk Between Tumor-Associated Microglia/Macrophages and CD8-Positive T Cells Plays a Key Role in Glioblastoma”. doi: 10.3389/fimmu.2021.650105. And this paper is more informative and discussed elaborative implications for therapeutic strategies in glioblastoma.
Author Response
Reviewer #2: I carefully evaluated the manuscript entitled “Divergent Crosstalk Between Microglia and T cells in Brain Cancers: Implications for Novel Therapeutic Strategies”. This paper can’t be accepted to publish in MDPI biomedines journal. Reasons for this decision:
Comments 1: The theme of this review paper “implications for novel therapeutic strategies” is missing. Authors are requested to discuss more on therapeutic implications.
Response 1: We acknowledge the importance of addressing the therapeutic implications of our findings, as it is a central theme of this review. In response to your request for further discussion on therapeutic relevance, we have enriched the manuscript by expanding the section on "Implications for Novel Therapeutic Strategies." This addition provides a more comprehensive exploration of how the insights presented in the review can inform and guide the development of innovative treatment approaches. We believe this revision significantly enhances the discussion of the therapeutic significance of our findings.
Comments 2: Authors are requested to add some infographics representing the theme of this review.
Response 2: Thank you for your valuable suggestion. In response to your request to include visual elements that represent the core themes of this review, we have added a graphical illustration to the manuscript. This infographic is designed to effectively summarize and highlight the key concepts and findings discussed, enhancing the clarity and accessibility of the review’s content.
Comments 3: Found similar review with title “Crosstalk Between Tumor-Associated Microglia/Macrophages and CD8-Positive T Cells Plays a Key Role in Glioblastoma”. doi: 10.3389/fimmu.2021.650105. And this paper is more informative and discussed elaborative implications for therapeutic strategies in glioblastoma.
Response 3: Thank you for your comment regarding the similarity of our review to previously published work on the interplay between tumor-associated microglia/macrophages and CD8+ T cells in glioblastoma. While we acknowledge the existence of similar reviews, we would like to emphasize that our manuscript expands its scope beyond glioblastoma by also including comprehensive information on the tumor microenvironment (TME) of metastatic brain tumors. Additionally, our review incorporates significant updates from studies published after 2021, offering the latest insights into the dynamic interactions within the TME. We believe these distinctions make our review unique and valuable, providing a broader and more current perspective on this important topic.
Round 2
Reviewer 1 Report
Comments and Suggestions for Authors
The authors have addressed my issues, I have no further questions.
Minor:
Line 366-369, the PMID should not appear in the manuscript as you have already cited these articles.
Author Response
Comments 1: Line 366-369, the PMID should not appear in the manuscript as you have already cited these articles.
Response 1: We have deleted the PMIDs. Thank you once again.
Reviewer 2 Report
Comments and Suggestions for Authors
The authors addressed all the queries and improved the manuscript significantly. So, it can be accepted in its present form to publish in MDPI biomedicines journal.
Author Response
Comments 1: The authors addressed all the queries and improved the manuscript significantly. So, it can be accepted in its present form to publish in MDPI biomedicines journal.
Response 1: Thank you for your comment!